# The Effects of Malonaldehyde on Quality Characteristics and Protein Oxidation of *Coregonus peled* (*Coregonuspeled*) during Storage

**DOI:** 10.3390/foods12040716

**Published:** 2023-02-07

**Authors:** Xin Guo, Na Wang, Yabo Wei, Pingping Liu, Xiaorong Deng, Yongdong Lei, Jian Zhang

**Affiliations:** 1School of Food Science and Technology, Shihezi University, Shihezi 832003, China; 2Key Laboratory for Processing and Quality Safety Control of Specialty Agricultural Products of Ministry of Agriculture and Rural Affairs (Provincial and Ministerial Cooperation), Shihezi University, Shihezi 832003, China; 3Key Laboratory for Food Nutrition and Safety Control of Xinjiang Production and Construction Corps, Shihezi University, Shihezi 832003, China

**Keywords:** *Coregonus peled*, storage, malondialdehyde, myofibrillar protein, quality

## Abstract

The effects of changes in the malondialdehyde (MDA) content on the quality of fish during the low-temperature storage period are unclear. Therefore, the effects of the MDA content on *Coregonus peled* quality and protein changes were investigated following storage under refrigeration (4 °C) and super chilling (−3 °C) for 15 days (d). The results showed that the MDA content continued to increase during storage and that the highest content was produced at 1.42 mg/kg during refrigeration. The fillet pH, drip loss, texture (hardness and elasticity), and myofibril fragmentation index deteriorated significantly during the storage period. Increased oxidation of the myofibrillar protein (MP) was observed in the 15 d storage period, and the MP carbonyl content was 1.19 times higher under refrigeration than in super chilling, while the protein α-helix structure decreased by 12.48% and 12.20% under refrigeration and super chilling, respectively. Electropherograms also showed that myosin degradation was particularly severe in the refrigeration storage period of 15 d. Overall, the MDA formed at the refrigeration and super chilling storage temperatures could promote structural changes in, and the oxidative degradation of, proteins to different degrees, leading to the deterioration of the fillet quality. This study provides a scientific basis for investigating the relationship between fish quality and changes in the MDA content during low-temperature storage.

## 1. Introduction

Fish, as one of the primary sources of high-quality protein for humans, are highly sought after by consumers. However, they are highly susceptible to spoilage, as they have high protein, fat, and moisture content [1]. Low-temperature storage is used as a traditional preservation method that reduces the extent of protein hydrolysis, microbial degradation, and lipid oxidation by endogenous enzymes during storage in order to delay the spoilage of the fish. The existing techniques mainly include refrigeration, ice storage, super chilling, and freezing. It is worth noting that super chilling ranges within 1–2 °C below the initial freezing point of fish [2]. This temperature not only prolongs the preservation time but also reduces the chances of muscle structure mechanical damage and protein quality deterioration, ensuring the quality of the fish.

Protein is the principal component of muscle tissue and is vital for various physiological and biochemical reactions related to fish softening and degradation. However, protein oxidation during storage is inevitable. Numerous studies have shown that lipid oxidation occurs before protein oxidation when lipids and proteins co-exist during storage [3,4]. Lipid oxidation products induce the formation of hydroxyl radicals, peroxyl radicals, and alkoxyl radicals that oxidate the proteins, resulting in the cross-linking and aggregation, structural disorders, and loss of functionality of proteins, which in turn affect the quality of the aquatic product [5,6]. Malondialdehyde (MDA) is the most representative secondary product of lipid oxidation, the content of which inevitably affects the process of protein oxidation.

*Coregonus peled* belongs to the family *Salmonidae*, *genus Coregonus* [7], and is a cold-water fish. In Xinjiang, China, *Coregonus peled* is regarded as a high-quality fish because the meat is tender, tasty, free of intermuscular spines, and rich in protein and unsaturated fatty acids [8]. However, it is mainly sold using cold chain methods due to the physiological characteristics of the fish upon death when emerging from the water. During cold chain transport or storage, the high-fat content of *Coregonus peled* can easily cause protein oxidation and reduce the quality of the fish. Thus, it is essential to understand the correlations between the degree of fat oxidation, changes in protein structure, and fish quality during storage. This study investigated the correlation between the changes in the MDA content of *Coregonus peled*, the changes in the fish meat texture, and the oxidative degradation of myofibrillar proteins, during refrigeration (4 °C) and super chilling (−3 °C) storage, with the aim of determining the preservation temperature of *Coregonus peled*.

## 2. Materials and Methods

### 2.1. Materials

Twenty fresh *Coregonus peled* (length 30 ± 2.5 cm; weight 1000 ± 50 g) were obtained from a commercial fishery processing company (Saihu Fishery Technology Development Co., Ltd., Wenquan Xian, China). After slaughter and washing with distilled water, the dorsal muscles of the fish were cut into uniform fillets (15 × 15 × 15 mm^3^). They were mixed well, packed into polyethylene plastic bags, stored in the refrigerator (BCD-212YMB/C, Rongsheng Refrigerator Co., Foshan, China) at −3 °C and 4 °C, and then collected at 0, 1, 3, 5, 7, 9, 11, 13, and 15 days within 3 replicates, respectively.

### 2.2. Determination of the MDA Content

According to Jung et al.’s [9] method, we detected the MDA content using thiobarbituric acid (TBA) combined with spectrophotometry, with slight modifications. The 50 mL mixture solution combined with trichloroacetic acid (TCA) (containing 0.1% ethylenediaminetetraacetic acid disodium salt (EDTA-Na_2_) and 7.5% TCA) was added to the sample (5 g) and homogenized. The mixture was placed in a stoppered conical flask and shaken at 50 °C for 30 min using a thermostatic oscillator (THZ-98, Shanghai Baidian Instruments Co., Shanghai, China). It was cooled to room temperature after the reaction was finished. The blend was filtered through double-layered filter paper, the initial filtrate was discarded, and the subsequent filtrate was reserved. The MDA standard (1 μg/mL) was diluted to 0.01–0.25 μg/mL using a TCA mixture. Filtrate and gradient dilution solutions (5 mL each) were transferred to a tube with a stopper, and the TCA mixture was regarded as the sample blank. Then, we added 5 mL of 0.02 mol/L TBA, mixed it well and sealed it, and then placed the reaction in a water bath at 90 °C for 30 min and cooled it with tap water. The absorbance was measured at 532 nm using a spectrophotometer (T3200, Shanghai YouKe Instrument Co., Ltd., Shanghai, China), and a standard curve was plotted for the diluted solution. 

### 2.3. Determination of pH

The pH of the fish fillets was tested with a pH meter (PHS-3E, INESA Scientific Instrument Co., Ltd., Shanghai, China) at room temperature on homogenates at 5000 rpm for 2 × 10 s in cooled distilled water at a ratio of 1:10 (*w*/*v*) using a homogenizer (S10- Scientz, Ningbo Xinzhi Biotechnology Co., Ningbo, China).

### 2.4. Drip Loss

The drip loss of the fillets was measured as described by Mu et al. [10]. After standing at room temperature for 20 min, we gently wiped away the water on the muscle surface with filter paper and then the samples were weighed again. The drip loss (%) was expressed as the ratio of the difference between the new weight (g) of the fillet and that of the initial weight of the fillet (g).

### 2.5. Texture Profile

Texture analysis was carried out according to the method of Qiu et al. [11] using a TA-XT2i texture analyzer (TA. XT Plus, Stable Micro Systems, Surrey, UK). Before the analysis, the samples (15 × 15 × 10 mm^3^) were equilibrated at room temperature. TPA testing was conducted with the flat-bottomed cylindrical probe P50 (stress deformation, 50.0%; trigger force, 10.0 g). For each sample, at least 3 determinations were obtained.

### 2.6. Myofibril Fragmentation Index (MFI)

The MFI was measured using the method of Liu et al. [12], with slight modifications. Each sample (2 g) was collected and homogenized using 40 mL of precooled extraction buffer (100 mmol/L KCl, 11.2 mmol/L K_2_HPO_4_, 8.8 mmol/L KH_2_PO_4_, 1 mmol/L ethylene glycol bis (2-aminoethyl ether) tetraacetic acid (EGTA), and 1 mmol/L MgCl_2_) for 1 min. The blend was centrifuged (Stratos, Thermo Fisher Technologies, Waltham, MA, USA) at 4000 rpm for 15 min at 4 °C, and the supernatant was discarded. The sediment was dissolved in the same buffer solution and centrifuged again. Lastly, the residue was dissolved to 0.5 ± 0.05 mg/mL in protein concentration using the same buffer. The suspension was measured at 540 nm using a UV spectrophotometer (T3200, Shanghai YouKe Instrument Co., Ltd., Shanghai, China). The MFI was calculated by multiplying the absorbance values by 200.

### 2.7. SEM Measurement

The method of Chen et al. [13] was employed with slight modifications. Small sample pieces (5 × 5 × 1 mm^3^) were cut from below the surface of the muscles. The cut fillets were diluted in 3% glutaraldehyde at 4 °C overnight. The treated samples were eluted with gradient (10%, 40%, 70%, and 100% ethanol) for 15 min, vacuum-dried for 24 h using a glass desiccator (diameter = 400 mm, Chengdu, China), and observed with scanning electron microscopy (SU8010, Hitachi, Chiyoda-ku, Japan).

### 2.8. Myofibrillar Protein (MP) Extraction

MP extraction was conducted as previously reported [14]. The samples (2 g) were homogenized with 30 mL of precooled, deionized water for 40 s. The mixture was centrifuged at 8000 rpm for 10 min at 4 °C. The residue was rehydrated in 0.3% NaCl and stirred for 15 min and then centrifuged again, in line with the initial step. By adding 0.02 mol/L Tris-HCl (containing 0.6 mol/L NaCl, pH 7.0) to the residue, we ensured that the mixture was mixed evenly for 1 h at 4 °C. The blend was filtered through gauze in two layers to eliminate insoluble or connective tissue. The filtrate was diluted with 4 times the volume of pre-cooled distilled water and centrifuged at 8000 rpm for 10 min at 4 °C. The sediment was MP, dissolved with buffer solution (0.6 mol/L NaCl), and the protein concentration was determined using the Biuret assay. Finally, the MP collected at different times was stored in a −80 °C freezer (DW-86L158, Zhejiang Jiesheng Refrigeration Technology Co., Huzhou, China). All the testing was completed within two weeks.

### 2.9. Carbonyl Content

The amount of carbonyl was determined based on 2,4-dinitrophenyl hydrazine (DNPH) derivatives, as described by Oliver et al. [15], with slight modification. The sample solution was mixed with 10 mmol/L of DNPH and reacted for 1 h in the dark at room temperature. Then, 30% TCA (final concentration) was added to stop the reaction. The residue was washed 3 times with ethyl acetate–ethanol (1:1) and dissolved with 6 mol/L guanidine hydrochloride for 15 min at 37 °C. The solution was centrifuged at 10,000 rpm for 5 min at 4 °C, and we then obtained the supernatant. The absorbance was measured at 370 nm. The carbonyl content was investigated using a molecular absorption coefficient of 22,000 M^−1^ cm^−1^ and measured in nmol/mg of protein.

### 2.10. Fourier-Transformed Infrared Spectroscopy (FTIR)

We performed an analysis of the protein secondary structure changes using a Fourier variation infrared spectrometer (Vertex 70v, Bruker, Karlsruhe, Germany). The sample processing method was based on Kang et al. [16]. The samples were treated with 100 times the volume for KBr compression, and spectra were recorded from 4000 to 400 cm^−1^. The protein secondary structure was analyzed using Peak Fit version 4.0 software and a Gaussian peak fitting algorithm.

### 2.11. Protein Solubility Measurement

As described by Chen et al. [17], we determined the protein solubility. The sample (2 mg/mL) was centrifuged at 4000 rpm and 4 °C for 15 min. The supernatant was collected after centrifugation, and the protein concentration was determined according to the Biuret method. Solubility (%) is expressed as the ratio of the supernatant to the original solution protein concentration.

### 2.12. Sodium Dodecyl Sulfate Polyacrylamide Gel Electrophoresis (SDS-PAGE)

SDS-PAGE was carried out to assess the molecular weight, according to He [18]. Protein samples were mixed at a concentration of 2 mg/mL in equal proportions to the loading buffer (0.5 mol/L Tris-HCl, pH 6.8; 20% SDS; 50% glycerol; 10% β-mercaptoethanol). The mixture was boiled at 95 °C for 5 min and stored at −80 °C before analysis. We used 5% polyacrylamide stacking gel and 12% polyacrylamide separating gel, which we added to the protein sample loaded with 10 μg, and a 10–250 kDa marker (Shanghai Epizyme Biomedical Technology Co., Ltd., Shanghai, China) was used to determine the molecular weight of the protein. A 1 L stock solution of electrophoresis buffer (30.3 g Tris, 144 g glycine, 10 g SDS) was prepared and diluted 10 times for the purpose of electrophoresis. After electrophoresis, the gels were stained with Coomassie Brilliant Blue R-250 and repeatedly decolorized with a decolorization solution (7.5% acetic acid and 25% methanol) until the bands were clear. Imaging analysis was carried out using a gel imaging system (Chemi Doc MP, Bio-Rad, Hercules, CA, USA).

### 2.13. Statistical Analysis

All data are presented in the form of means ± standard deviations (SD). The significance of the data was assessed with a one-way analysis of variance (ANOVA) and analyzed with Duncan’s method using SPSS 17.0 to compare the differences between data. The significance level was set at *p* < 0.05. The correlation was estimated with Pearson’s correlation coefficient. All the experiments were performed in triplicate.

## 3. Results and Discussion

### 3.1. MDA Content

The unsaturated fatty acids in *Coregonus peled* are prone to oxidation. MDA, as the main product of lipid oxidation, has very high oxidation activity, and the oxidation of protein leads to the deterioration of fish quality.

As seen in Figure 1A, the MDA content increased in proportion to the storage time. The unsaturated fatty acids were oxidized, and the MDA content increased with the change in temperature and time spent during storage. During refrigeration, the content of MDA increased slowly from 0 to 9 days, but after 9 days, the MDA content increased sharply. The same changes were observed during super chilling. However, after day 3, the MDA content was significantly higher during refrigeration than during super chilling (*p* < 0.05). The MDA content under refrigeration was 1.29 times higher than that under super chilling on day 15. This phenomenon may be explained by the fact that the temperature of 4 °C increased the lipid oxidase activity and consumed the unsaturated fatty acids more rapidly than −3 °C, thus producing lipid oxidation by-products [19].

### 3.2. pH

As shown in Figure 1B, the pH of the fillet during storage decreased under the two storage conditions, and the trend was more evident during refrigeration (*p* < 0.05). The pH decreased significantly (*p* < 0.05) in the early stages of super chilling storage from 1 to 5 d. On day 13, the pH increased, but there was no significant difference (*p* > 0.05). On the one hand, oxygen is no longer provided to the fish body after slaughter, and glycolysis produces lactic acid, pyruvic acid, and other substances. On the other hand, phosphoric acid production and the induction of ATP and protease can slowly decrease the pH value of fish [20]. In this study, overall, the pH showed a decreasing trend, probably because the alkaline substances produced by protein degradation were insufficient to neutralize the acidic substances produced by glycogen degradation. In the later storage stage, proteins and some nitrogen compounds are degraded into polypeptides and amino acids under the joint actions of enzymes and microbial proliferation, releasing alkaline substances and increasing the pH [21]. The pH values decreased significantly (*p* < 0.05) when the fish were refrigerated. The pH fell to 95.70% of the initial value on day 9 and increased on days 9–11, but it continued to decrease after day 11 because the higher storage temperature promoted endogenous enzyme activity in the fish muscle and strengthened the microbial action. The pH values under super chilling were higher than those under refrigeration. The pH values were significantly different (*p* < 0.05) at 9 d under refrigeration and 11 d under super chilling, indicating that super chilling storage can effectively slow down the deterioration of fish and help to prolong the preservation period.

### 3.3. Drip Loss

Changes in the fish drip loss during the different temperature storage conditions are shown in Figure 1C. As seen in Figure 1C, the drip loss increased significantly at both storage temperatures with increasing storage time (*p* < 0.05). It is worth noting that the fish drip loss stored under super chilling was higher than that under refrigeration (*p* < 0.01). In particular, on day 15, the drip loss of the fish under super chilling was 1.50 times higher than that observed under refrigeration. When stored under super chilling, the rate of drip loss increased rapidly within 5 d, and the drip loss was more significant. However, the increased rate of drip loss tends to reduce in the later storage time, due to mechanical damage to the cell wall caused by ice crystal formation during the freezing process and the juice exudation, preventing migration back into the tissue cells during the thawing process [22].

### 3.4. Texture 

After slaughter, fish are stiff and softened, and their texture changes during storage, which can reflect the meat quality. Table 1 shows that the hardness and springiness declined significantly with the prolongation in the storage time (*p* < 0.05). After 15 d of storage under refrigeration and super chilling, the hardness declined by 68.21% and 66.26%, and the springiness decreased by 48.91% and 39.13%, respectively, compared to the initial values. In contrast, the fillets were harder after being stored under super chilling, which can be attributed to the higher water loss from the fillet tissue. Textural changes are associated with water loss, microbial metabolism, and the autolysis of enzymes in carbohydrates, proteins, and lipids [23]. The deterioration of the fish texture over time indicated that the internal fibers of the fish lost their original form, the Z-lines of the myofiber gradually became blurred and fragmented, and the connective tissue was damaged, ultimately leading to protein breakdown [24]. There was a more significant difference (*p* < 0.01) in the springiness and hardness between the different temperature storage conditions. In addition, the changes in hardness and springiness were more minor under super chilling, indicating that storage under super chilling can effectively delay meat deterioration and maintain a better texture of the fillets.

### 3.5. Myofibril Fragmentation Index (MFI)

The MFI reflects the integrity of myofibrils and skeleton proteins, which effectively characterize the quality of the fish. The MFI value is directly proportional to the degree of myofibril damage [25]. 

As seen in Figure 1D, the MFI values increased significantly with storage time (*p* < 0.05). The MFI value under refrigeration was higher than that under super chilling during the storage process, but there was no significant difference (*p* > 0.05). After storage under refrigeration for 5 d, the MFI increased sharply with the storage time until 13 d. In contrast, the MFI rose sharply in the first 5 days of super chilling storage. After storage for 15 d, the MFI values under refrigeration and super chilling were 4.24 and 4.08 times the initial values, respectively. The MFI may be related to the structural integrity of the protein. MP degradation during storage affects the dissociation of the myogenic fibril structure. Jiang et al. showed that the degradation of skeleton proteins containing titin, nebulin, and troponin-T might be the reason for the increase in the MFI [26].

### 3.6. Microstructure

Figure 2 and Figure 3 show the changes in muscle microstructure over 15 days under super chilling and refrigeration storage conditions, respectively. The fish tissue structure and integrity gradually became damaged with the extended storage time. During refrigeration for 3 d, the muscle surface gradually loosened, pores appeared, and the muscle fibers became deformed and broken. After storage for 5 d, the space of the muscle cut surface had become increasingly larger, and the muscle fiber was no longer compact and even, indicating that the myofibrils had been gradually destroyed [27]. After 11 d of storage, granular material appeared on the surface of the muscle fibers, which can mainly be regarded as connective tissue, such as collagen and specific sarcoplasmic proteins [28]. The boundary between the muscle fibers became blurred, indicating that the intima muscle had begun to rupture. However, no apparent changes occurred during super chilling for 5 d, and the muscle cut surface was smooth with slight cracks. After storage for 11 d, the structure became loose and uneven. The muscle fibers appeared disorderly, and the pores became larger. This might be primarily related to the loss of water in the muscles. The water between the muscle bundles condenses into ice during super chilling, which increases the volume and compresses the internal muscle fibers, resulting in structural changes [29]. The microstructure changes that occurred during super chilling storage were significantly less marked than those observed during refrigeration, implying that the lower storage temperature was beneficial for protecting muscle integrity.

### 3.7. Carbonyl Content

The change in protein carbonyl content is often regarded as a significant indicator of protein oxidation, which measures the protein oxidation degree. Figure 4A demonstrates that the carbonyl content showed an upward trend with storage time, indicating that the oxidative degeneration of MP occurred during storage. After 15 days of super chilling storage and refrigeration, the MP carbonyl values increased by 119.23% and 83.80%, respectively, compared to day 0. During the storage process, carbonyl compounds were mainly generated through the oxidization of amino acids on the protein skeleton’s side chain, peptide backbone rupture, and molecular rearrangement [29]. Moreover, the MDA molecules double bond reacted with the protein active group (histidine residues of the imidazole group, lysine residues of the epsilon-amino group, and cysteine residues of the thiol-reactive group, 2019) by nucleophilic addition, promoting the formation of a Schiff base, which lead to protein carbonylation [30]. In the same storage period, the protein carbonyl values under super chilling were consistently lower than those under refrigeration. In particular, after day 9, highly significant differences (*p* < 0.01) were observed, demonstrating that super chilling effectively reduced protein oxidation compared with refrigeration.

### 3.8. Secondary Structure

FTIR is the primary method used to reflect protein secondary structure. Figure 4C,D shows the changes in the MP infrared spectra of the fillets stored under refrigeration and super chilling for 15 d, respectively. There were apparent characteristic absorption peaks at 1700–1600 cm^−1^ (Amide I) and 1600–1500 cm^−1^ (Amide II), of which the former reflects chemical bond changes in C=O tensile vibration, C-N telescopic vibration, and N-H bending vibration. The latter primarily reflects chemical bond changes in primarily N-H bending and C-N and C-C stretching [31]. It is worth noting that the absorption peaks at 1246.56 cm^−1^ and 1047.02 cm^−1^ were more pronounced only in the initial spectra, which may have been caused by changes in the telescopic vibration of C-H during storage [32]. 

Changes in the MP secondary structure content were obtained after the amide I band was resolved. Table 2 and Table 3 show the relative content of the MP secondary structure of the fillets stored under refrigeration and super chilling for 15 d, respectively. After 15 days of storage under refrigeration and super chilling, the contents of α-helix declined by 12.48% and 12.20%, and the β-sheet rose by 23.33% and 21.69%, respectively. Some studies have reported that a decrease in the α-helix was accompanied by an increase in the β-sheet, usually due to protein molecular denaturation and unfolding [33,34]. However, the contents of β-turn and random coil fluctuated and changed slightly. The mentioned results showed that during storage, MP oxidation caused the protein molecules to unfold and the partial hydrogen bonds to weaken, destroying the ordered structure of the α-helix and transforming it into a β-sheet and random coil.

### 3.9. Solubility

Solubility can reflect changes in protein cross-linking and aggregation. As shown in Figure 4B, the MP solubility decreased gradually with the extension of storage time. The MP solubility decreased by 18.83% and 24.48% after 15 d of storage under refrigeration and super chilling, respectively. The MP solubility decreased during storage, which might be due to changes in the internal protein structure, such as disulfide and hydrogen bonds, leading to the increased cross-linking of molecules and the formation of insoluble aggregates affecting protein solubility [35]. In addition, the formation of ice crystals leads to the loss of intracellular water flow outside the cell and further reduces protein solubility. However, refrigeration and super chilling had no significant effect (*p* > 0.05) on the change in MP solubility during the same storage period.

### 3.10. SDS-PAGE

The gel electrophoresis patterns under refrigeration and super chilling can be observed in Figure 5, respectively. The sample densities, indicated by the corresponding band color, showed slight differences between refrigeration and super chilling storage for 15 d, as displayed on the electrophoresis gel. In the 0 d sample, myosin heavy chain (MHC) occurred mainly in the complete form. MHC gradually degraded and formed band 1 and band 2 with the prolonged storage time, in which degradation bands clearly appeared after 5 d of storage under refrigeration, while they appeared slightly after 9 d under super chilling storage. Troponin-T showed a similar trend. The troponin-T band tended to blur, and the degraded band 3 (27–30 kDa) appeared with the prolongation of storage time. In addition, myosin light chain (MLC-1, MLC-2) and troponin-C were degraded to varying degrees during storage, but the degradation was more severe under refrigeration. All these results indicated that MP was degraded during storage, and the effect was more significant under refrigeration than super chilling. MHC, MLC-1, and MLC-2 changed significantly because myosin is sensitive to free radicals during storage [36]. In the case of the present results, the MDA content generated by refrigeration storage for 15 d was 1.29 times higher than that observed under super chilling, which may account for the more pronounced degradation of myosin during refrigeration. In addition to free radical factors, endogenous enzymes may contribute to protein degradation. Endogenous enzymes may hydrolyze MP, but low temperatures inhibit endogenous enzyme activity [37]. Therefore, MP was less degraded under super chilling than under refrigeration.

### 3.11. Correlation Analysis of the MDA Content with Other Indicators

Figure 6 depicts the correlation between the changes in the MDA content and the fillet quality and texture, protein oxidation, and secondary structure during storage. The MDA content demonstrated a significant linear correlation with the abovementioned indicators under two conditions (4 °C and −3 °C), in which the MDA content showed a significant negative correlation (*p* < 0.01) with the muscle pH, texture characteristics (hardness and elasticity), MP solubility, and α-helix, while the drip loss, MFI, carbonyl, and β-sheet showed a significant positive correlation (*p* < 0.01) with the MDA content. In addition, the MDA content was negatively correlated with β-turn (*p* < 0.05) and had a significant positive correlation with random curl (*p* < 0.01) during 4 °C storage. However, the MDA content was not significantly different from β-turn (p > 0.05) and had a significant negative correlation with random curl (*p* < 0.01) during −3 °C storage. 

Increased drip loss in stored fillets is a well-established fact of post-slaughter degradation [38]. During storage, lipid oxidation gradually occurred with time and the change in temperature, thus increasing the MDA content. It has been reported that MDA is susceptible to addiction or cycloaddition reactions with nucleophilic reagents in low pH environments, producing various condensation products [39]. More importantly, lipid oxidation (MDA) induces protein oxidation. The oxidation of structural proteins, including myofibrillar protein and connective tissue, leads to the disruption of myofibril and skeletal protein integrity, a decrease in textural properties, and muscle tissue sparing [4,40]. All of the abovementioned results indicate that the MDA content is closely related to changes in fish storage quality.

## 4. Conclusions

The fillet quality decreased during storage, and the 4 °C condition was not conducive to storing the fillets compared with −3 °C. During 15 d storage at 4 °C and −3 °C, the hardness and springiness of the fillets tended to decrease, and the muscle microstructure changed from tight to loosen. The quality indicators (pH, drip loss, and MFI) deteriorated with the prolongation of storage time. Additionally, the protein structure was destroyed by the increased MP carbonyl content and solubility. The MP secondary structure appeared to change, and MP degradation occurred in the case of MHC, MLC, and troponin. The results of the correlation analysis between the MDA content and the abovementioned indicators showed that the increased MDA content significantly affected the fillet quality and MP oxidative degradation during storage. 

## Figures and Tables

**Figure 1 foods-12-00716-f001:**
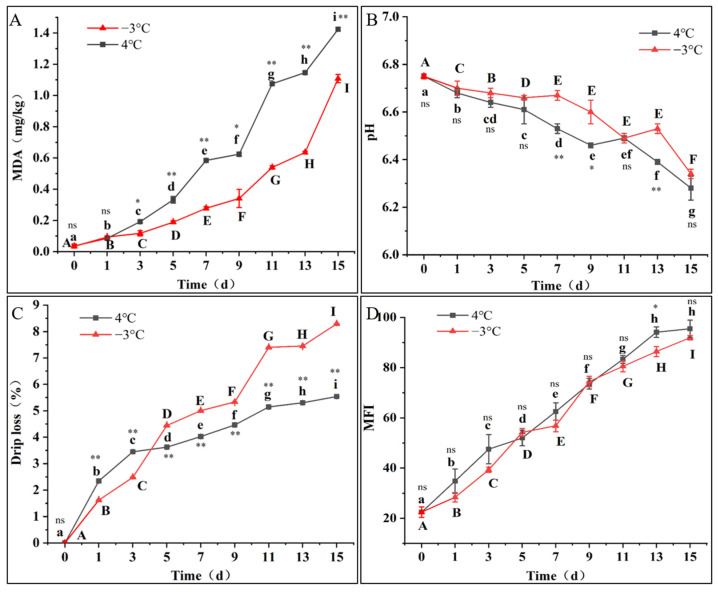
Changes in MDA content (**A**), pH (**B**), drip loss (**C**), and MFI (**D**) of *Coregonus peled* fillets during 4 °C refrigeration and −3 °C super chilling storage for 15 d. Note: The different letters represent a significant difference between samples by time storage (*p* < 0.05); different uppercase letters (A–I) indicate significant differences (*p* < 0.05) for −3 °C super chilling. Different lowercase letters (a–i) indicate significant differences (*p* < 0.05) for 4 °C refrigeration. Significantly different values as influenced by storage temperature: * (*p* < 0.05); ** (*p* < 0.01); ns: no significant difference (*p* > 0.05). Specific numerical data are in the Appendix A.

**Figure 2 foods-12-00716-f002:**
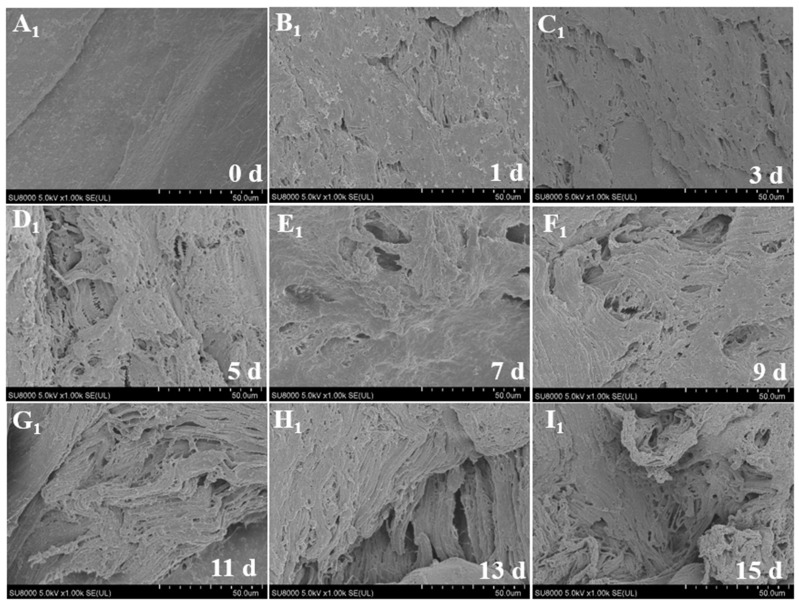
Changes in microstructure of *Coregonus peled* fillets under 4 °C refrigeration storage for 15 d. (**A_1_**–**I_1_**) indicate storage in 4 °C refrigeration for 0–15 d.

**Figure 3 foods-12-00716-f003:**
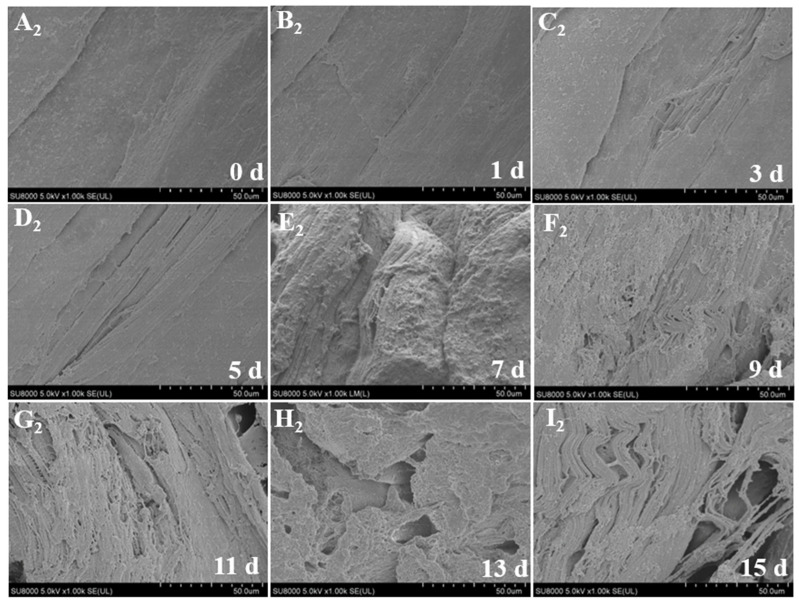
Changes in microstructure of *Coregonus peled* fillets under −3 °C super chilling storage for 15 d. (**A_2_**–**I_2_**) indicate storage in –3 °C super chilling for 0–15 d.

**Figure 4 foods-12-00716-f004:**
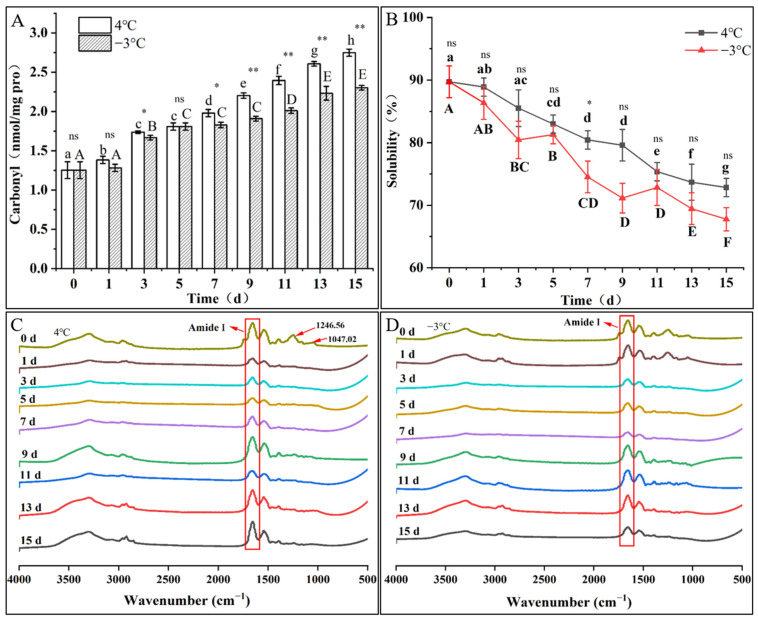
Myofibrillar proteins changes in carbonyl (**A**), solubility (**B**), and Fourier infrared spectroscopy (**C**,**D**) during 4 °C refrigeration and −3 °C super chilling storage for 15 d. Note: The different letters represent a significant difference between samples by storage time (*p* < 0.05); different uppercase letters (A–F) indicate significant differences (*p* < 0.05) for −3 °C super chilling. Different lowercase letters (a–h) indicate significant differences (*p* < 0.05) for 4 °C refrigeration. Significantly different as influenced by storage temperature: * (*p* < 0.05); ** (*p* < 0.01); ns: no significant difference (*p* > 0.05). Specific numerical data about carbonyl and solubility are in the Appendix A.

**Figure 5 foods-12-00716-f005:**
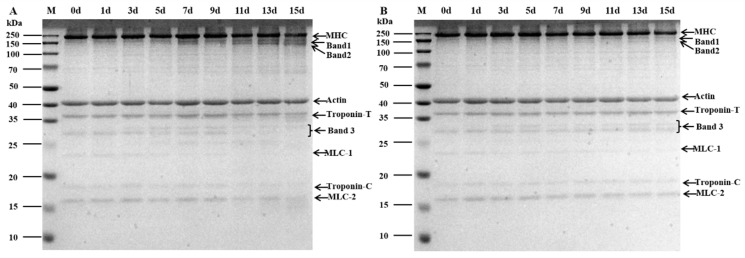
Myofibrillar proteins changes in SDS-PAGE of *Coregonus peled* fillets during 4 °C refrigeration (**A**) and −3 °C super chilling (**B**) storage. M: marker.

**Figure 6 foods-12-00716-f006:**
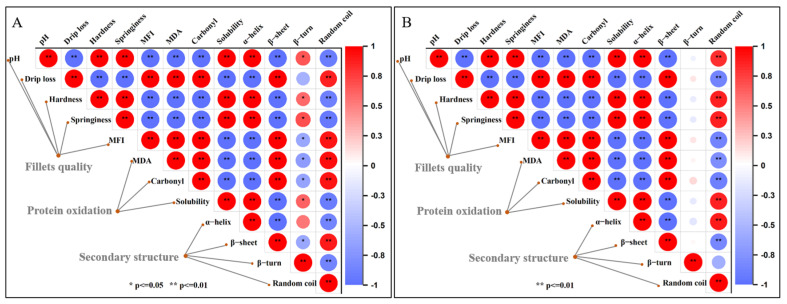
Clustering heatmap based on Pearson’s correlation tests between fillet quality, protein oxidation, and secondary structure of *Coregonus peled* under 4 °C refrigeration (**A**) and −3 °C super chilling (**B**) storage. Darker red and darker blue indicate higher positive and negative correlations.

**Table 1 foods-12-00716-t001:** Changes in the texture of *Coregonus peled* fillets stored at 4 °C and −3 °C for up to 15 days.

Time	Hardness (g)		t	Springiness		t
d	4 °C	−3 °C		4 °C	−3 °C	
0	1119.7 ± 39.3 i	1119.7 ± 39.3 i	ns	0.92 ± 0.06 i	0.92 ± 0.06 i	ns
1	1067.9 ± 10.0 h	992.3 ± 55.9 h	ns	0.89 ± 0.04 h	0.87 ± 0.16 h	ns
3	965.3 ± 10.0 g	913.8 ± 3.3 g	**	0.86 ± 0.05 g	0.82 ± 0.06 g	**
5	734.9 ± 7.2 f	817.3 ± 21.6 f	**	0.83 ± 0.10 f	0.79 ± 0.07 f	**
7	608.6 ± 6.4 e	762.2 ± 34.7 e	**	0.75 ± 0.12 e	0.76 ± 0.01 e	ns
9	545.4 ± 2.4 d	656.9 ± 11.7 d	**	0.69 ± 0.07 d	0.71 ± 0.01 d	ns
11	464.4 ± 10.2 c	570.6 ± 22.2 c	**	0.62 ± 0.07 c	0.66 ± 0.09 c	**
13	418.1 ± 6.4 b	464.8 ± 9.9 b	**	0.51 ± 0.16 b	0.60 ± 0.01 b	**
15	355.9 ± 3.3 a	377.8 ± 8.4 a	*	0.47 ± 0.15 a	0.56 ± 0.02 a	**

Data are the mean ± standard deviation of three replicates. Values in the same column with different lowercase letters (a–i) are significantly different by storage time (*p* < 0.05). t: significantly different as influenced by storage temperature: * (*p* < 0.05); ** (*p* < 0.01); ns: no significant difference (*p* > 0.05).

**Table 2 foods-12-00716-t002:** Relative contents of secondary structures in *Coregonus peled* fillets stored at 4 °C for 15 days.

Time (d)	α-Helix (%)	β-Sheet (%)	β-Turn (%)	Random Coil (%)
0	35.41 ± 0.04 ^a^	19.50 ± 0.02 ^a^	30.11 ± 0.03 ^c^	14.98 ± 0.01 ^e^
1	34.03 ± 0.03 ^b^	21.42 ± 0.02 ^c^	30.34 ± 0.04 ^d^	14.21 ± 0.02 ^de^
3	33.54 ± 0.02 ^c^	21.33 ± 0.02 ^b^	31.05 ± 0.03 ^g^	14.08 ± 0.02 ^d^
5	32.57 ± 0.03 ^d^	21.97 ± 0.02 ^d^	30.90 ± 0.03 ^f^	14.56 ± 0.02 ^h^
7	31.90 ± 0.01 ^e^	22.87 ± 0.03 ^f^	30.50 ± 0.01 ^e^	15.54 ± 0.03 ^g^
9	31.49 ± 0.03 ^f^	22.71 ± 0.02 ^e^	31.19 ± 0.02 ^h^	14.61 ± 0.02 ^f^
11	31.09 ± 0.03 ^g^	22.82 ± 0.02 ^f^	30.97 ± 0.07 ^f^	15.12 ± 0.05 ^c^
13	31.00 ± 0.03 ^h^	23.77 ± 0.02 ^g^	29.91 ± 0.02 ^b^	15.32 ± 0.02 ^b^
15	30.99 ± 0.07 ^h^	24.05 ± 0.05 ^h^	29.45 ± 0.02 ^a^	15.51 ± 0.01 ^a^

Data are the mean ± standard deviation of three replicates. Values in the same column with different lowercase letters (a–h) are significantly different (*p* < 0.05).

**Table 3 foods-12-00716-t003:** Relative contents of secondary structures in *Coregonus peled* fillets stored at −3 °C for 15 days.

Time (d)	α-Helix (%)	β-Sheet (%)	β-Turn (%)	Random Coil (%)
0	35.41 ± 0.04 ^a^	19.50 ± 0.03 ^a^	30.11 ± 0.04 ^b^	14.98 ± 0.01 ^g^
1	34.34 ± 0.02 ^b^	19.44 ± 0.01 ^a^	30.94 ± 0.01 ^f^	15.28 ± 0.02 ^h^
3	33.45 ± 0.02 ^c^	20.03 ± 0.03 ^b^	32.41 ± 0.03 ^i^	14.11 ± 0.01 ^b^
5	33.43 ± 0.04 ^d^	21.10 ± 0.01 ^c^	31.27 ± 0.03 ^g^	14.20 ± 0.03 ^c^
7	33.04 ± 0.05 ^e^	22.67 ± 0.03 ^e^	29.82 ± 0.04 ^a^	14.47 ± 0.05 ^e^
9	32.43 ± 0.04 ^f^	22.41 ± 0.07 ^d^	30.87 ± 0.05 ^e^	14.34 ± 0.04 ^d^
11	32.29 ± 0.02 ^g^	22.61 ± 0.03 ^e^	30.55 ± 0.01 ^c^	14.55 ± 0.03 ^e^
13	31.52 ± 0.03 ^h^	23.18 ± 0.02 ^f^	31.35 ± 0.01 ^h^	13.95 ± 0.02 ^a^
15	31.09 ± 0.04 ^i^	23.73 ± 0.03 ^f^	30.75 ± 0.06 ^d^	14.43 ± 0.04 ^f^

Data are the mean ± standard deviation of three replicates. Values in the same column with different lowercase letters (a–i) are significantly different (*p* < 0.05).

## Data Availability

The data presented in this study are available on request from the corresponding author.

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
