# Peer review of "The Effects of Malonaldehyde on Quality Characteristics and Protein Oxidation of Coregonus peled (Coregonuspeled) during Storage"

_foods, 2023, doi:10.3390/foods12040716_

Round 1

Reviewer 1 Report

Molecules           23.1.2023

Title: it is preferred to replace the common name with scientific name or add it besides the scientific name

In general, the manuscript don’t follow the style of FOODs, the authors need to consider that in revision

Abstract:

The first three lines are confused, the sentence is not complete, check

Clear the aim of the work and add tentative line at the debut of abstract

Language is very bad and incomprehensible, check the abstract and hole manuscript for linguistically and structural errors, make sentences are simple and comprehensible

Rewrite the abstract with numerical data, clear the aim and add recommendation at the end of abstract

Add the study model in keywords

Line 60, no need for italic

The authors should clear at the end of introduction the novelty and the hypothesis behind conducting this research is to search for a new preservation method or to determine the temperature

Line 87 EDTA sodium salt

Clear the sample size

Provide the origin and model of all devices

In SDS PAGE, clear the gel concentration, marker used, the buffer system, what do you use to identify protein bands, such as gel documentation

Enhance the statical analysis part and clear the pre-tests (homogeneity,) used to analyze your data with ANOVA

Table 1 indicate the lowercase letters in the table footnotes and do that in all tables

Line 292, Figure

Reformulate the results presentation, don’t repeat the values form tables or figures

Enhance figure 2

Enhance electropherogram images

Enhance and reduce conclusion it is general, be concentrate

References style needs to be reformulated following the journal instruction and check out the outputs of all references

Reviewer 2 Report

The article entitled "The effects of malonaldehyde on quality characteristics and protein oxidation of Coregonus Peled during storage" by Gup et al. evaluates the quality of Coregonus Peled meat stored at 4 and ‑3 ºC over time (15 days) and compares it with the malonaldehyde (MDA) content. At the same time, the research compares the two different conservation methods (at 4 and -3 ºC). I believe that the article should attend to major corrections.

Major comments

The comparison between the two conservation methods carried out throughout the entire manuscript is statistically invalid, since the analysis of variance (ANOVA) has not been carried out for it, or at least said analysis is not reflected anywhere (P value not indicated, neither in the figures/ tables nor in the text) (e.g., lines 204-205; 209-203; 260-261; 330-333, etc.). I request that the differences between both storage temperatures be analyzed for each day, thus being able to discuss the results on a statistical basis. To do this, the results must be displayed in a better way. I propose to show them in tables where the P for the time is collected (analysis already shown in the figures) and the P for the storage temperature.

 Days

1

2

3

4

5

6

7

8

9

10

11

12

13

14

15

t

MDA (mg/ kg)

Refrigeration

Super chilling

T

pH

Refrigeration

Super chilling

T

t: significantly different values as influenced by time storage: *(P<0.05); **(P<0.01); ***(P<0.001); ns:  no significant difference.  T: significantly different values as influenced by storage temperature; *(P<0.05); **(P<0.01); ***(P<0.001); ns: no significant difference. x–y Means within the same row not followed by the same letter differ significantly (influence of storage time) (P<0.05).

Minor comments

Abstract

Lines 16-32: The "Abstract" section does not provide a summary of the article. A background is not included, nor are the methods carried out for its conduction specified. As indicated in the guide for authors of Molecules, the "Abstract" should have a 1) Background; 2) Methods; 3) Results; and 4) Conclusions, (not including headings). Rephrase the abstract accordingly.

Keywords

Line 33: Remove the capital "S" from the word "storage".

Introduction

Line 37: "Traditional preservation method" instead of " Traditional storage method " to avoid being repetitive.

Line 60: Remove the italics from the words that should not carry it.

Line 61: This sentence is confusing, the Coregonus peled has not become a cold-water fish. You must rewrite the sentence so that it indicates that it is a cold-water fish and later that it has become a high-quality fish.

Material and Methods

Line 75: Use the italic font to write Coregonus peled. Check the rest of the manuscript to confirm that the scientific name is always written in italics.

Line 84: It is the first time you name "TBA" in the main text, you must explain the meaning of this abbreviation, [e.g., thiobarbituric acid (TBA)].

Line 85: Consider noting the full name "ethylenediaminetetraacetic acid" before the acronym. The "2" of "Na2" must be indicated as a subscript.

Line 109: Add a space between "20" and "min".

Line 124: It is the first time you name "EGTA" in the main text, you must explain the meaning of this abbreviation.

Line 152: "Min" instead of "minutes".

Line 160: The meaning of the abbreviation "2,4-dinitrophenyl hydrazine" has been explained in previous lines. Remove "2,4-dinitrophenyl hydrazine" and keep only DNPH.

Line 182: It is the first time you name "SDS-PAGE" in the main text, you must explain the meaning of this abbreviation, [e.g., sodium dodecyl sulfate polyacrylamide gel electrophoresis (SDS-PAGE)].

Line 192-196: Nothing is explained about how the determination of the correlations were carried out. Add this information, please.

Results and Discussion

Line 215: You should explain the difference between uppercase and lowercase letters (a‑i for refrigeration and A‑I for super chilling).

Line 273: Please review the units in the table. Generally, the hardness is expressed in N, and the springiness in mm.

Line 288: Add the reference year in parentheses.

Line 340: The meaning of the abbreviation "FTIR" has been explained in previous lines. Remove "2,4-dinitrophenyl hydrazine" and keep only DNPH.

Line 344: Write "-1" as superscript. Please review the entire document accordingly.

Line 356: Add a comma before "respectively".

Line 390: Add a space between "band" and "1".

Line 394:"kDa" the "k" in lower case.

Round 2

Reviewer 2 Report

Thanks for following the suggestions I made. I believe that the document now offers a better discussion of the findings achieved about the topic of study. I appreciate the effort and the kind responses of the authors to my questions.